# CRISPRs in the human genome are differentially expressed between malignant and normal adjacent to tumor tissue

Job van Riet [1,2,3,13], Chinmoy Saha [4,13], Nikolaos Strepis[4], Rutger W. W. Brouwer[5],
Elena S. Martens-Uzunova [1], Wesley S. van de Geer [2,3], Sigrid M. A. Swagemakers [6], Andrew Stubbs[6],
Yassir Halimi [4], Sanne Voogd [4], Arif Mohammad Tanmoy [4,7], Malgorzata A. Komor[8,9],
Youri Hoogstrate [10], Bart Janssen[11], Remond J. A. Fijneman [8], Yashar S. Niknafs[12], Arul M. Chinnaiyan [12],
Wilfred F. J. van IJcken [5], Peter J. van der Spek [6], Guido Jenster [1] & Rogier Louwen [4✉]

Clustered Regularly Interspaced Short Palindromic Repeats (CRISPRs) have been identified in bacteria, archaea and mitochondria of plants, but not in eukaryotes. Here, we report the discovery of 12,572 putative CRISPRs randomly distributed across the human chromosomes, which we termed hCRISPRs. By using available transcriptome datasets, we demonstrate that hCRISPRs are distinctively expressed as small non-coding RNAs (sncRNAs) in cell lines and human tissues. Moreover, expression patterns thereof enabled us to distinguish normal from malignant tissues. In prostate cancer, we confirmed the differential hCRISPR expression between normal adjacent and malignant primary prostate tissue by RT-qPCR and demonstrate that the SHERLOCK and DETECTR dipstick tools are suitable to detect these sncRNAs. We anticipate that the discovery of CRISPRs in the human genome can be further exploited for diagnostic purposes in cancer and other medical conditions, which certainly will lead to the development of point-of-care tests based on the differential expression of the hCRISPRs.

[1] Department of Urology, Erasmus MC Cancer Institute, University Medical Center Rotterdam, Rotterdam, Netherlands. [2] Cancer Computational Biology Center, Erasmus MC Cancer Institute, University Medical Center Rotterdam, Rotterdam, Netherlands. [3] Department of Medical Oncology, Erasmus MC Cancer Institute, University Medical Center Rotterdam, Rotterdam, Netherlands. [4] Department of Medical Microbiology and Infectious Diseases, Erasmus University Medical Center Rotterdam, Rotterdam, Netherlands. [5] Center for Biomics, Erasmus University Medical Center Rotterdam, Rotterdam, Netherlands. [6] Clinical Bioinformatics, Department of Pathology, Erasmus University Medical Center Rotterdam, Rotterdam, Netherlands. [7] Child Health Research Foundation, 23/2 SEL Huq Skypark, Block-B, Khilji Rd, Dhaka 1207, Bangladesh. [8] Translational Gastrointestinal Oncology, Department of Pathology, Netherlands Cancer Institute, Amsterdam, Netherlands. [9] Oncoproteomics Laboratory, Department of Medical Oncology, VU University Medical Center, Amsterdam, Netherlands. [10] Department of Neurology, Erasmus University Medical Center Rotterdam, Rotterdam, Netherlands. [11] GenomeScan, Leiden, Netherlands. [12] Michigan Center for Translational Pathology, University of Michigan, Ann Arbor, MI, USA. [13] These authors contributed equally: Job van Riet, Chinmoy Saha. ✉email: r.louwen@erasmusmc.nl

In 1987, researchers identified a new class of repeats in the genome of the prokaryote *Escherichia coli* and named them interspaced Short Sequence Repeats[1]. Five years later, the same type of repeats where discovered in the prokaryote *Mycobacterium tuberculosis*[2], whereas other researchers identified them in the archaea *Haloferax volcanii* and *Haloferax mediterranei*[3]. Soon thereafter, it became apparent that these types of repeats, renamed as Short Regularly Spaced Repeats (SRSR), were widely distributed across bacterial and archaeal genomes and were even identified in the mitochondria of plants[4,5]. The SRSR repeats are short, usually 24–40 base pairs (bp) in length, harbor a unique and recognizable layout characterized by inner and terminal inverted repeats of up to 11 bp that allow the formation of palindromes[4]. The palindromic repeats are separated by variable sequences, named spacers and commonly are between 20 and 58 bp in length[4]. SRSR elements generally appear in clusters, although isolated variants harboring only two repeats with a single spacer were identified as well[4,5]. A high sequence homogeneity for SRSR elements is found between taxa, however, specifically in archaea, a more heterogenic architecture is reported, with repeats harboring less than 85% of sequence identity[4]. This specific architecture distinguishes SRSR elements from other repeat forms, such as tandem or the interspersed repeats[6,7]. In prokaryotes, the size of SRSR elements ranges from 70 bp up to several kilobases in length[3,4]. A potential role in endogenous gene expression was proposed for bacterial SRSR elements[6], whereas others found the repeat architecture to be useful for bacterial species identification purposes[2]. More functional-related studies suggest that the SRSR elements may play a role in chromosomal replication in archaea[3], possibly via a centromere-like function[8]. Simply spoken, fulfilling a role as anchors to separate the chromosomes from each other. Evidence was also provided that the archaeal SRSR elements could be bound by DNA-binding proteins[9], or were shown to be actively expressed in both the *Euryarchaeota* and *Crenarchaeota*[10], the latter of which is closely related to the eukaryotes[11].

With a rapidly increasing number of publications a problem arose with the SRSR-related literature nomenclature, since authors used also terminology, such as Short Sequence Repeats[1], Tandem REPeats[3], Direct Variant Repeats[12], Large Clusters of 20 bp Tandem Repeat sequences[13], or SPacers Interspersed Direct Repeats[5], to name a few. Therefore, in 2002, Mojica et al., and Jansen et al., discussed and agreed upon to rename this specific repeat family to Clustered Regularly Interspaced Short Palindromic Repeats (CRISPR)[6,14]. At that time Jansen et al., also reported that the CRISPR arrays were often accompanied with a specific set of genes harboring a nuclease function, which were called CRISPR-associated genes (*cas*)[14]. Today, this complete system is known as CRISPR-Cas, in prokaryotes a well-defined system shown to play roles in the biological arms-race with foreign genetic elements[15], endogenous gene regulation[16], and virulence features[17]. Nonetheless, it has been thought that such a system does not exist in eukaryotes and viruses[14].

Despite these beliefs, CRISPR-Cas elements were discovered in the genomes of bacteriophages[18] and in a giant virus[19]. At that time, these discoveries made us question the scientific consensus that such systems are absent from the eukaryotic genomes[3–6,14] (Supplementary Note 1). Here, we address the presence of putative CRISPR-like elements in the human genome and transcriptome, which we termed hCRISPRs. We applied an in silico pipeline to analyze several human transcriptome datasets and show that the hCRISPRs are actively transcribed as small noncoding RNAs (sncRNAs) in cell lines and human tissues. Finally, by using RT-qPCR we confirm the differential expression of hCRISPRs in prostate cancer and demonstrate a proof-of-principal to detect hCRISPRs by making use of point-of-care test methodologies.

## Results

### Identification and characterization of hCRISPRs.
To investigate the presence of hCRISPRs in the human genome and transcriptome, we developed an in silico CRISPR-Cas identification pipeline, that utilizes the detection tool CRISPRCasFinder[20] (Fig. 1). Using our pipeline, we interrogated the human reference genome (GRCh38.p13) and identified 12,572 hCRISPRs (Figs. 1, 2a and Supplementary Data 1). The identified hCRISPRs harbor one or more variable sequences (spacers) separated by short regular repeats (Supplementary Fig. 1 and Supplementary Data 1). Furthermore, the hCRISPRs harbor inner inverted and terminal repeats enabling the formation of a palindrome (Supplementary Fig. 1), a unique finding reported before, when the CRISPRs were identified in prokaryotes[4,21–23]. The hCRISPRs are found across all chromosomes and their distribution is directly correlated to the chromosome length ($R^2 = 0.94$; Supplementary Fig. 2a, b). The vast majority of 12,046 hCRISPRs contain only two repeats and one spacer, whereas 526 hCRISPRs are clustered with three or more repeats separated by spacer sequences (Supplementary Data 1). We further established that the average genomic size of the hCRISPRs is 105 bp in length (Supplementary Fig. 2c), while their overall size ranges from 59 to 1638 bp (Supplementary Data 1). The average size of the direct repeats is 30 bp in length (Supplementary Fig. 2d). The hCRISPRs thus share remarkable similarities with their prokaryotic counterparts both in repeat size, spacer presence, and architecture[4], which we further investigated in more detail (Supplementary Note 2 and Supplementary Fig. 3a, b, c and Supplementary Data 2, 3, 4). From these analyses we concluded that CRISPRCasFinder identified by far the biggest number of hCRISPRs with consensus repeats that shared the strongest similarities to the consensus repeats found within the well-known prokaryotic CRISPR-Cas systems.

This prompted us to further characterize them for functionalities within the human genome by overlapping the hCRISPR genomic locations (start to end) with annotations of known genes, promotors and transcripts, obtained from GENCODE v33 (GRCh38)[24]. We discovered that a substantial portion ($n = 7403$; 58%) overlaps with transcripts and genes with a diverse range of genetic and molecular purposes (Fig. 2b). Most of these genes are classified as protein coding ($n = 3689$), pseudogenes ($n = 280$), long noncoding RNAs, or other RNA transcripts ($n = 1643$) (Fig. 2b). Additionally, 6420 hCRISPRs overlap with intronic regions of annotated genes, while 455 overlap with distal intergenic regions (Fig. 2c). The remaining hCRISPRs ($n = 528$) overlap with exons, promoters, or untranslated regions (Fig. 2c). We then compared the genomic positions of the hCRISPRs against known human repeat regions and discovered that 4425 (35%) of them could not be linked to any known human repeat family (Fig. 2d, Supplementary Data 1). In contrast, 8147 hCRISPRs (65%) were found to overlap known human repeat sequences (Fig. 2d, Supplementary Data 1). For example, we observed that retroviral-related human repeat families, such as MER11C, LTR12E, LTR12C, and HERVH-int, often overlap or harbor embedded hCRISPRs (Fig. 2e, Supplementary Data 1).

### hCRISPRs are actively expressed as sncRNAs in human cell lines.
In archaea, CRISPR arrays are known to be expressed as sncRNAs[10]. To address whether the hCRISPRs are also actively expressed and maintained as sncRNAs, we took advantage of available noncoding RNA-sequencing datasets generated from the human cell lines K562, HeLa S3, Hep-G2, and HU-V-ECC for ENCODE[25]. These datasets allowed us to determine that the transcription of the hCRISPRs was distinct and could be detected in each of the cell lines analyzed (Fig. 3a, b and Supplementary Data 1).

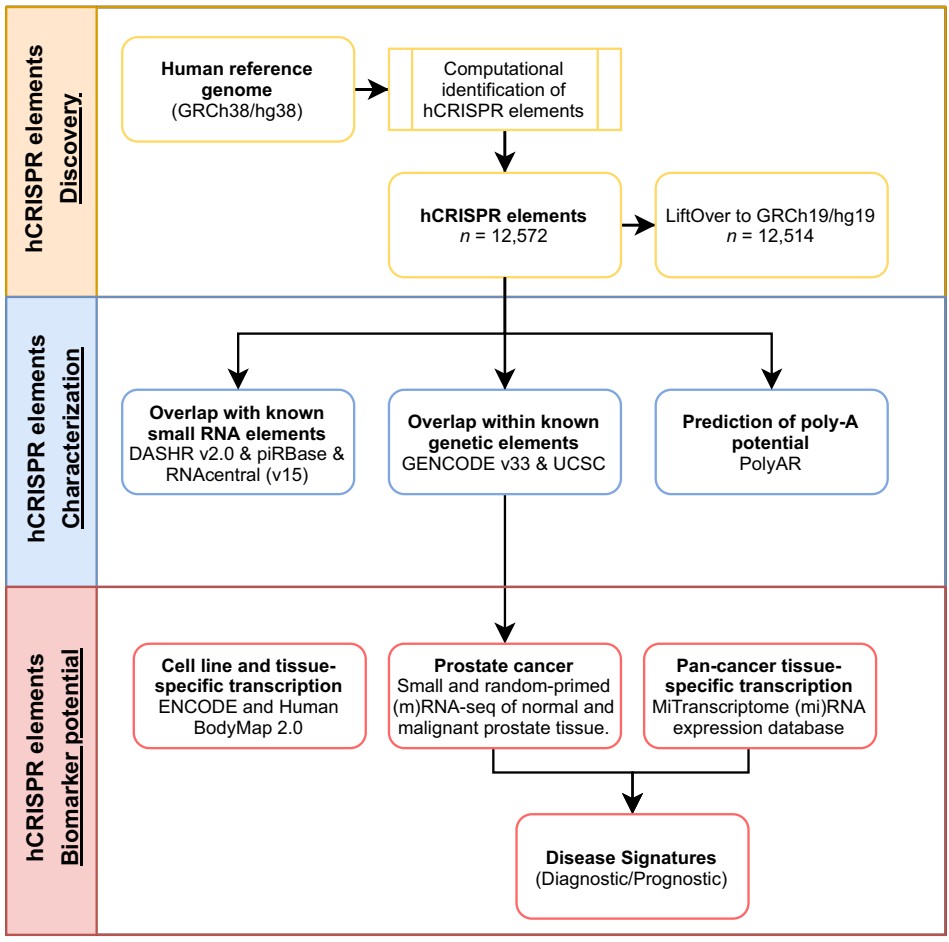

**Fig. 1 Overview of study design to detect and characterize the hCRISPRs.** Schematic flowchart highlighting the major steps of the in silico approach as employed in this study. The major steps can be grouped as the following; identification of CRISPRs in the human reference genome (GRCh38); genomic characterization of the identified hCRISPRs ($n = 12,572$); expression and the discovery of a diagnostic potential for the hCRISPRs in cell lines, distinct human tissues (Illumina Bodymap 2.0 and DASHR 2.0), prostate cancer (NGS-ProToCol), and a pan-cancer (localized) cohort (MiTranscriptome).

In archaea, CRISPR arrays are recognized by multiple DNA-binding proteins with unknown function[9]. Transcription factors are a class of proteins that also recognizes a specific DNA context and regions on the human genome, regulating DNA transcription to RNA[26]. We therefore wondered whether the hCRISPRs resided within known transcription factor binding sites (TFBS), enabling the binding of transcription factors[26]. In addition, the observed expression of hCRISPRs suggests their presence in open chromatin regions/structures that can be revealed by DNase I activity and can be affected by methylation. For that purpose, the genomic positions of TFBS, DNase I hypersensitive sites, and CpG methylated genomic islands were obtained from UCSC genome browser[27], which we overlapped with the genomic positions of the hCRISPRs. These analyses demonstrated that 4126 of the 12,572 hCRISPRs (33%) overlap with one or more TFBS positions (Fig. 3c and Supplementary Data 1 and 5), 2668 (21%, $p$-value < 0.001, Z-score: 4.9507) with DNase I hypersensitive sites (Fig. 3d and Supplementary Data 1) and 104 (1%, $p$-value: 0.15, Z-score: 1.1334) with CpG methylated islands (Fig. 3d and Supplementary Data 1). Interestingly, we noticed an enrichment for footprints of DNA-directed RNA polymerase II subunit RPB1 (POLR2A) (5.5%, $p$-value < 0.001, Z-score: 7.079) that overlap with 696 hCRISPRs (Fig. 3c and Supplementary Data 1 and 5). RPB1 is a RNA polymerase II that transcribes genes encoding messenger RNAs, many functional noncoding RNAs and is encoded by the gene *POLR2A*[28–30]. This made us wonder whether the hCRISPRs harbored poly-A tail recognition and cleavage signatures, which we inspected with PolyAR[31]. Our PolyAR analysis revealed that 5231 of the hCRISPRs (42%) harbor one or multiple poly-A recognition sites (Supplementary Data 1).

The hCRISPRs genomic coordinates were also overlapped with noncoding RNAs from DASHR v2.0[32], piRBase[33], and RNAcentral[34] (release 15) resources. These three resources capture a wide range of known RNA classes including (among others): long noncoding RNAs; sncRNAs; ribosomal RNAs; PIWI-interacting RNAs; microRNAs; signal recognition particle RNAs; transfer RNAs; and small nuclear and nucleolar RNAs. We observed that 9012 distinct hCRISPRs (72%) overlap (minimum of 5 bp) with at least one of the RNA classes present in the combined resources; most prominently with long noncoding RNAs (and/or another RNA class; $n = 7872$ (63%)) and PIWI-interacting RNAs (and/or another RNA class; $n = 2393$ (19%)) (Fig. 3e and Supplementary Data 1). However, the vast majority of long noncoding RNA-overlapping hCRISPRs ($n = 7103$; 90%) overlaps (on average) with less than one percent of the total size of their superimposed long noncoding RNA (mean width: 24,307 bp), which is not unexpected concerning the longer length of the long noncoding RNAs compared to the hCRISPRs. Conversely, hCRISPRs that overlap with PIWI-interacting RNAs ($n = 2393$) are, on average, 5.8 times larger when compared to their superimposed PIWI-interacting RNA (mean width: 28 bp).

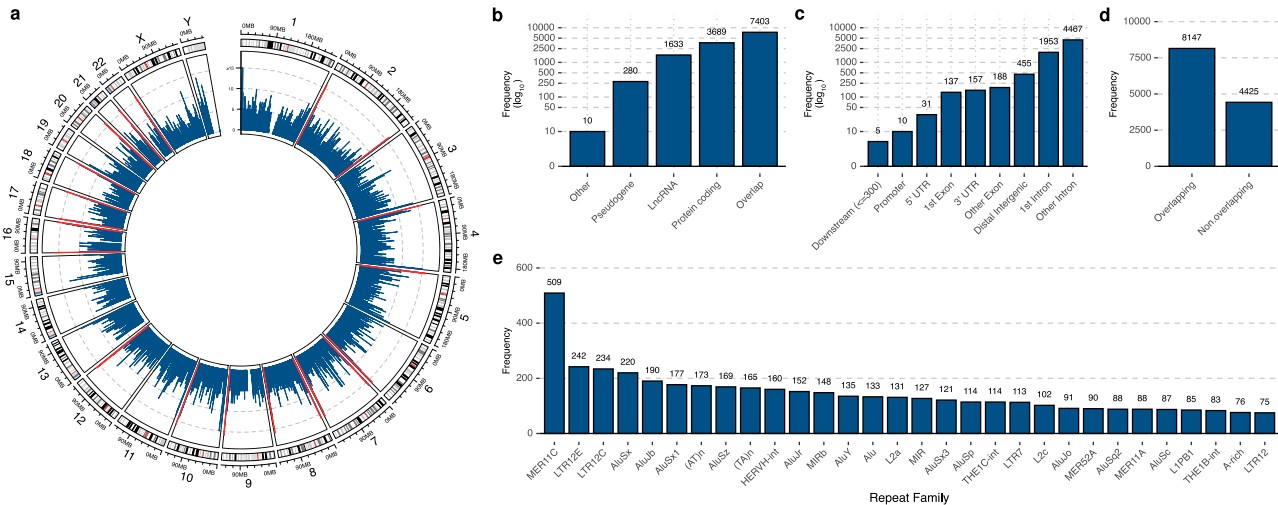

**Fig. 2 Genomic representation of the hCRISPRs. a** Genomic location of the hCRISPRs ($n = 12,572$) visualized as an ideogram, each window represents a human chromosome (chromosome 1:22 and XY). The $y$-axis of the data-tracks shows the number of hCRISPRs, binned per genomic Mb (in blue; bins with >15 hCRISPR are colored red). **b** Overlap of the hCRISPRs with known annotations from GENCODE v33 (transcript support level 1 and 2). Overlap was based on a minimum of five base pairs between hCRISPR foci and gene foci; several hCRISPRs overlap with >1 gene. Frequencies ($x$-axis) are plotted on a $\log_{10}$ scale. **c** Overlap of the hCRISPRs with known genetic features from GENCODE v33 (transcript support level 1 and 2). Overlap was based on a minimum of five base pairs between hCRISPR foci and gene foci; several hCRISPRs overlap with >1 gene. Frequencies ($x$-axis) are plotted on a $\log_{10}$ scale. **d** Overlap of the hCRISPRs with known human repeat families. Overlap was based on a minimum of five base pairs between hCRISPR foci and repeat foci; several hCRISPRs overlap with >1 repeat. **e** Top 30 (based on frequency) of repeat families overlapping with the hCRISPRs.

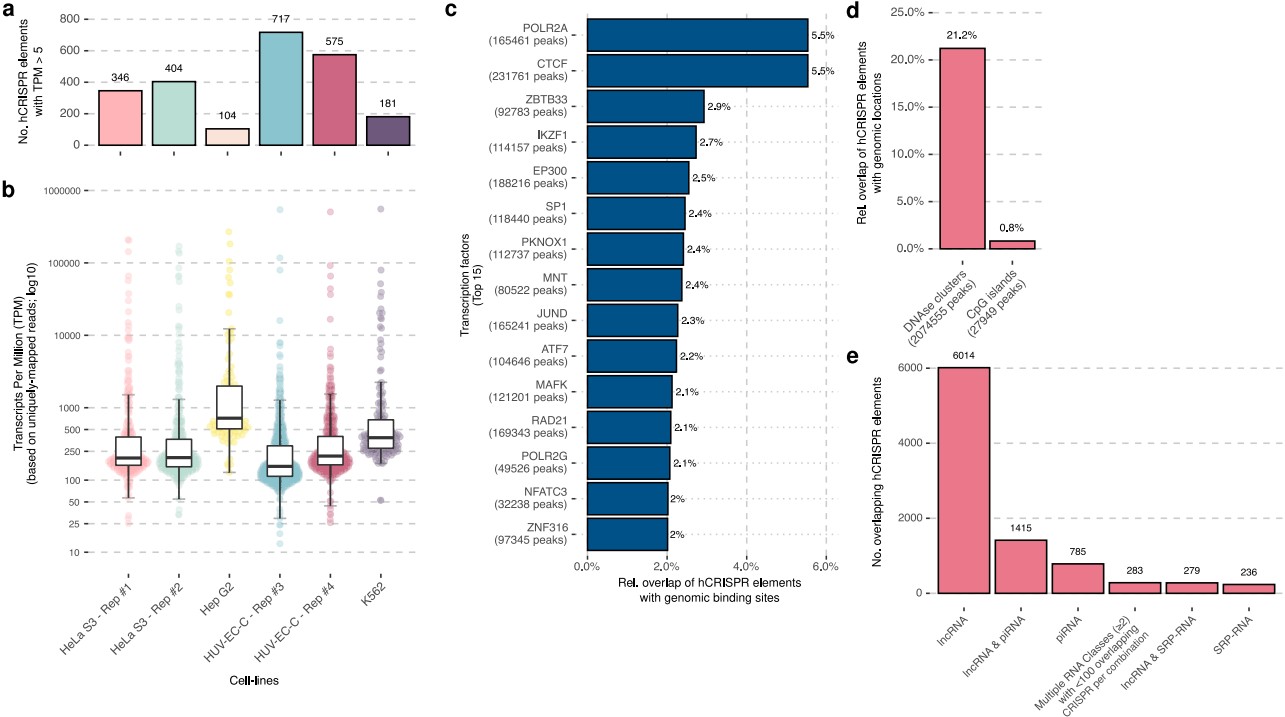

**Fig. 3 A subset of the hCRISPRs is actively transcribed in human cell lines and reside within TFBS. a** Number of hCRISPRs, with ≥5 TPM in four distinct RNA-sequenced human cell lines (HeLa S3 and HUVEC-C both have two replicates). **b** Boxplots representing the TPM of the hCRISPRs in four distinct RNA-sequenced human cell lines (HeLa S3 and HUVEC-C both have two replicates); median, Q1 and Q3 are highlighted with a bold black line and error bars, respectively. The $y$-axis is displayed in $\log_{10}$-scale. **c** Overlap of the hCRISPRs with TFBS of 161 transcription factors from ENCODE. Overlap was based on a minimum of five base pairs between hCRISPR foci and binding site foci; the 15 most-overlapped TFBS with the hCRISPRs are shown. The $y$-axis depicts relative frequencies (number of overlaps divided by total number of hCRISPRs). **d** Overlap of the hCRISPRs with DNase I clusters from 125 cell types (right) and CpG islands (left) was based on a minimum of five base pairs. **e** Overlap of the hCRISPRs with known annotations of small RNA from DASHR 2.0, piRBase, and RNAcentral (v15).

This reveals that hCRISPRs do not overlap (exactly) with other known RNA classes. Furthermore, 3560 (28%) of them do not overlap (even partially) with any of the known RNA classes at all.

**hCRISPRs are actively expressed in a wide variety of human cell lines and tissues.** The active transcription of hCRISPRs in the human cell lines K562, HeLa S3, Hep-G2, and HU-V-ECC made us further explore their expression by using the DASHR 2.0 database (GRCh38)[32]. This database harbors the normalized expression of sncRNA transcripts (≤100 bp) from a wide variety of cell types, cell lines and tissues ($n = 185$)[32]. Analysis of raw datasets from the four different databases that together form DASHR 2.0[32], led to the discovery that a substantial number of the DASHR 2.0 sncRNAs, originate from the hCRISPRs. In DASHR1 GEO Hg38 this number was 3600 sncRNAs; in DASHR 2 GEO Hg38, this number was 5573 sncRNAs; in Encodedata-portal Hg38, this number was 4495 sncRNAs; and in Encode GEO Hg38 this number was 2104 sncRNAs (Supplementary Data 6). Eight hundred ninety sncRNAs that originate from hCRISPRs were detected in all four databases (Supplementary Data 6). Further inspection of the obtained DASHR 2.0 data revealed that 884 unannotated DASHR 2.0 sncRNAs are consistently detected across different human tissues and cell types and originate from our discovered elements (Supplementary Data 7). A total of 392 of these DASHR 2.0 sncRNA transcripts even harbor a specific annotation after they were analyzed in a sequencing-based pipeline for analysis of sncRNAs, named SPAR[35] (Supplementary Data 8). Additionally, by making use of the DASHR v2.0 UCSC Genome browser hub[32], we observed that the DASHR 2.0 sncRNA transcripts that originate from hCRISPRs overlapped with coding or noncoding regions of the human genome (Supplementary Data 6, 7 and 8). 337 of the DASHR 2.0 sncRNA transcripts that originate from the hCRISPRs and have a SPAR annotation had a tissue-specific expression (Q score[32] was <7) (Supplementary Data 8). Some of these hCRISPR transcripts were identified in multiple experimental replicates (Supplementary Data 7 and 8), indicating that these findings are reproducible, even between different research groups. The expression of hCRISPRs was further validated with RNA-sequencing datasets from different tissues types obtained from the Illumina Human BodyMap 2.0 project[36,37] (Fig. 4a, b). Indeed, this analysis established that the expression of the hCRISPRs was tissue-type-specific (Fig. 4c), with only a few of them being expressed across multiple tissue types ($n = 153$), whereas 5104 of the hCRISPRs (41%) were not expressed at all in any of the tissue types analyzed (Fig. 4a–d and Supplementary Data 1).

**Diagnostic potential of RNA transcripts overlapping hCRISPRs.** Previously, we analyzed the sncRNA content in malignant prostate tissues obtained from tumors at different disease stages and with different Gleason grades[38,39] (GEO-id GSE80400). This dataset is also included in the DASHR 2.0 database[32] (Supplementary Data 7 and 8). Upon reanalysis, a total of 29 additional hCRISPRs were discovered as differentially expressed between malignant and normal adjacent to tumor prostate tissue ($p \leq 0.05$) (Supplementary Data 1). These additional hCRISPRs are different than the nine previously identified transcripts[38,39]. From these 38 (29 + 9) differentially expressed hCRISPRs, 30 were predominantly expressed (≥2× fold-change) in malignant prostate tissue, while elevated expression of eight others was mainly detected in normal adjacent to tumor tissue. This finding suggests that characteristic hCRISPR patterns might be useful to distinguish malignant from non-malignant tissue or, more broadly stated, stratify healthy individuals and patients.

In the publicly available whole-transcriptome sequencing dataset (NGS-ProToCol EGAS00001002816)[40–42], which includes 50 malignant and 40 normal adjacent to tumor prostate tissues, 177 hCRISPRs revealed a substantial difference in expression levels between malignant and normal adjacent to tumor prostate tissue (Fig. 5a, b, c and Supplementary Data 1). Out of these, 103 were found to be upregulated while 74 were downregulated in malignant prostate tissue; 10 differentially expressed hCRISPRs are highlighted in Fig. 5d. Additionally, previous research on prostate cancer-specific biomarkers revealed a set of transcripts known as the Erasmus MC prostate cancer-associated transcripts[43]. From this work, it became apparent that several Erasmus MC prostate cancer-associated transcripts were overlapping or originated from the hCRISPRs ($n = 182$; 1.5%) (Supplementary Data 1), re-enforcing their potential as putative biomarkers.

To determine the pan-cancer biomarker potential of the hCRISPRs, we took advantage of the MiTranscriptome datasets generated by large-scale RNA-sequencing[44]. After filtering the available MiTranscriptome cohorts ($n = 70$) by removing cohorts with fewer than 10 normal and 10 cancer samples, we were left with 12 extensive cancer-cohorts that we used for further differential analyses (Fig. 6a and Supplementary Data 9). When MiTranscriptome transcript coordinates were overlaid with hCRISPR coordinates (lifted over to GRCh37), 10,373 out of 12,572 hCRISPRs (82.5%) overlapped with 9361 distinct transcripts (Supplementary Data 1 and 9). Of these, 4622 distinct transcripts were differentially expressed ($\log_2$ fold-change ≥ 1, avg. read-count ≥ 10 within the respective cohort and $q \leq 0.01$) in one or multiple cohorts (Fig. 6b, c and Supplementary Data 1 and 9); a single differential expressed hCRISPR per cohort is shown in Fig. 6d. A subset of these MiTranscriptome transcripts ($n = 581$) was differentially expressed in only one cohort (cohort-specific) (Fig. 6c and Supplementary Data 1). Interestingly, out of the 177 hCRISPRs differentially expressed between malignant and normal adjacent to tumor prostate tissue within the NGS-ProToCol dataset, 104 revealed similar patterns within the MiTranscriptome dataset and 10 of these were also previously identified as Erasmus MC prostate cancer-associated transcripts[43] (Supplementary Data 1).

Next to RNA-sequencing technologies, microarrays have been used extensively to study the role of RNA transcripts in disease development[45]. Related to our findings that sncRNAs were originating from our hCRISPRs and could be used to distinguish malignant from normal adjacent to tumor prostate tissue, the Affymetrix U133 plus 2 microarray became of interest to us, since this array contains probes to detect not only the expression of coding but also sncRNAs[45,46]. By making use of the genomic positions of the U133 plus 2 probe regions, individual probes and the hCRISPRS (lifted over to GRCh37), we were able to identify 4440 overlapping probe regions with 71 distinct U133 plus 2 sncRNA probes that exactly matched the genomic position of some of the hCRISPRs (Supplementary Data 1). Analyzing the expression profiles of these Affymetrix probes revealed that the sncRNAs, resembling transcripts that originated from a hCRISPR ($n = 71$), allowed us to differentiate between healthy individuals and patients that were battling cancer, autoimmune diseases, sepsis, or harbored fertility problems among others (Supplementary Fig. 4 and Supplementary Data 10 and 11).

To further validate the expression of hCRISPRs within our interrogated samples, by an independent technique, we designed quantitative real-time PCR (qPCR) LNA-assays for two hCRISPRs that showed robustness in multiple published prostate cancer datasets containing sncRNAs, namely chr9_209 and chr19_106 and tested these in a panel of matched tissue and prostate cancer samples (part of the NGS-ProToCol dataset). For

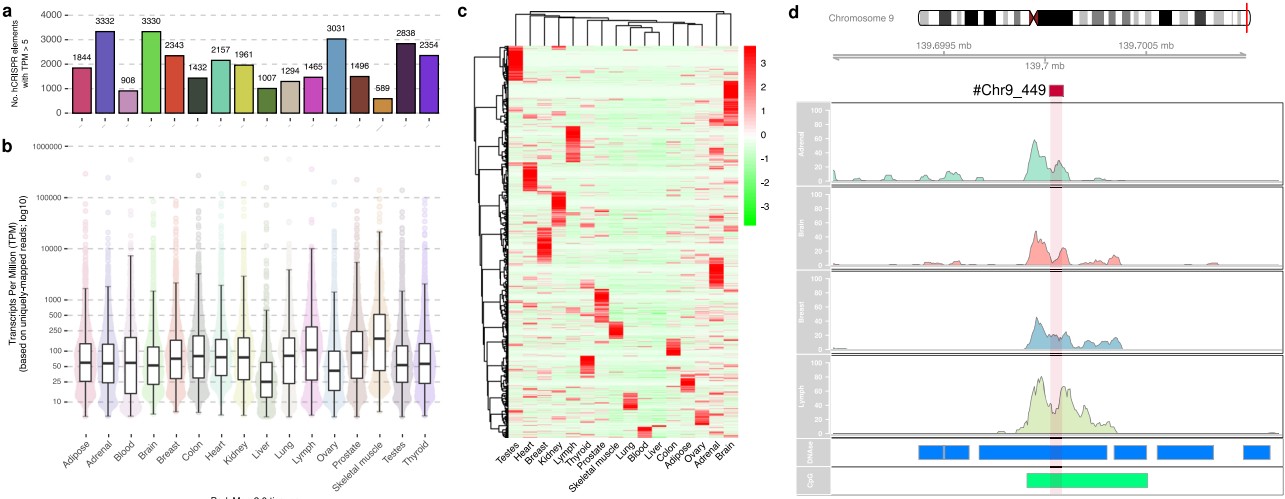

**Fig. 4 A subset of the hCRISPRs is actively transcribed in healthy human tissues. a** Number of hCRISPRs, with ≥5 TPM per tissue in 16 distinct RNA-sequenced human tissues from the Illumina BodyMap 2.0 cohort. **b** Boxplots representing the TPM of the hCRISPRs in 16 distinct RNA-sequenced human tissues from the Illumina BodyMap 2.0 cohort; median, Q1 and Q3 are highlighted with a bold black line and error bars, respectively. The y-axis is displayed in log_{10}-scale. **c** Unsupervised clustering (Euclidean; Ward.D2) of TPM (as Z-scores) of all hCRISPRs with ≥100 TPM in at least one BodyMap tissue (n = 3258). Negative Z-scores are highlighted in green whilst positive Z-scores are highlighted in red colors. **d** Example of a hCRISPR (#chr9_449 located at chr9:136805479-136805563 on hg19) which overlaps with a known human gene and shows transcriptional activity within several human BodyMap 2.0 tissues. Upper tracks display the ideogram (hg19) and genomic location with ticks per 5000 bp, the third to seventh track display the expression profile (number of primary-aligned reads) per tissue (Adrenal gland, Brain, Breast, and Lymph Node). Lower tracks display overlapping DNase I clusters (in blue) and CpG islands (in green).

each patient, a tumor sample containing at least 70% cancer cells and a normal adjacent to tumor sample counterpart were analyzed. Their differential expression of both elements was confirmed by us using small-RNA-Seq[38,39], chr9_209 being upregulated and chr19_106 being downregulated in malignant prostate tissue. The qPCR results confirmed the concordant presence of 5'-phosphorylated sncRNA transcripts corresponding to the identified hCRISPRs chr9_209 and chr19_106. Furthermore, the expression patterns measured by qPCR followed those as measured by RNA-Seq (Fig. 7a and Supplementary Fig. 5a, b, c and Supplementary Table 1 and Supplementary Data 12). We observed distinct upregulation of chr9_209 in malignant tissue in three out of four patients (1, 2, and 4) and similar low expression levels in patient 3 as observed by whole-transcriptome sequencing. Likewise, chr19_106 was downregulated in malignant tissue in three out of four cases, similar to the expression detected by whole-transcriptome sequencing; only patient 2 revealed an unexpected opposite expression pattern.

We then explored the possibility to detect the hCRISPR RNAs of chr9_209 and chr19_106 by using the SHERLOCK[47] and DETECTR[48] dipstick technologies. After optimization, the DETECTR and SHERLOCK technology enabled the detection of chr19_106 (Fig. 7b) and chr9_209 (Fig. 7c), respectively.

## Discussion

CRISPR arrays have been reported in bacteria, archaea, and in the mitochondria of plants[3–6], but were thought to be absent from the genome in eukaryotes[5]. In this work we thus successfully challenged the established conceptual idea of eukaryotic genomes lacking CRISPR-Cas systems (see also Supplementary Note 3). Indeed, our finding is supported by recent observations of CRISPR-Cas systems in the eukaryotic genome[49,50]. From an evolutionary perspective, some interesting questions can be asked, as our phylum analyses revealed that the hCRISPRs could be assigned to at least 12 different phyla, in which the kingdoms *Metazoa* (animals), *Paramavirae* (retroviruses), and *Viridiplantea*

(green plants) were overrepresented. CRISPR-Cas systems are mobile and new insights on the evolution of CRISPR-Cas indicate that orphan CRISPRs and *cas* genes are actually two separated systems that, by coincidence, have found each other by horizontal gene transfer and evolved in some prokaryotes to the defense system currently known as CRISPR-Cas[51–53]. The kingdom of life exists of three domains, namely Archaea, Bacteria, and Eukarya, which all can be infected by the mobile and reverse transcribing *Paramavirae*[54]. Our finding that 39% of the hCRISPRs are accompanied by the reverse transcriptase RVT_1 thus makes us wonder whether retroviruses or retrotransposons are one of the drivers behind the spread of CRISPR-Cas across the domains of life (see also Supplementary Note 3).

Next, also from a functional perspective, intriguing questions can be asked, for example, related to our discovery that 33% of the hCRISPRs overlapped with known TFBS, and 65% did this with known repetitive sequences. Such a link between repetitive DNA and TFBS is not new as is the embedding of these regulatory elements in transposons; both findings have been reported before[55,56]. Indeed, TFBS arising from endogenous retroviral elements, the *Paramavirae* group, were of the most conserved type and are found in transposon related elements[54,55]. This type is also suggested to play important roles in regulatory functions, for example primate-specific ERV related TFBS are controlled by TP53[57], of which the latter, when not working properly, increases the risk for certain cancers[58]. Another important finding is, that TFBS act cell-type-specific, which in case of the hCRISPRs applies to their expression, a coincidence? Probably not, since the hCRISPRs positions in the human genome were found to be enriched with the footprints for RPB1, a subunit of DNA-directed RNA polymerase II, a more general transcription factor that is encoded by *POLR2A*[56]. RPB1 synthesizes mRNA precursors, including many functional noncoding RNAs[28,29] and thus might be a driver behind the expression of sncRNAs originating from the hCRISPRs. In addition, we also want to mention that CTCF was enriched in connection to the hCRISPRs. CTCF is involved in many cellular processes, including transcriptional regulation, V(D)J

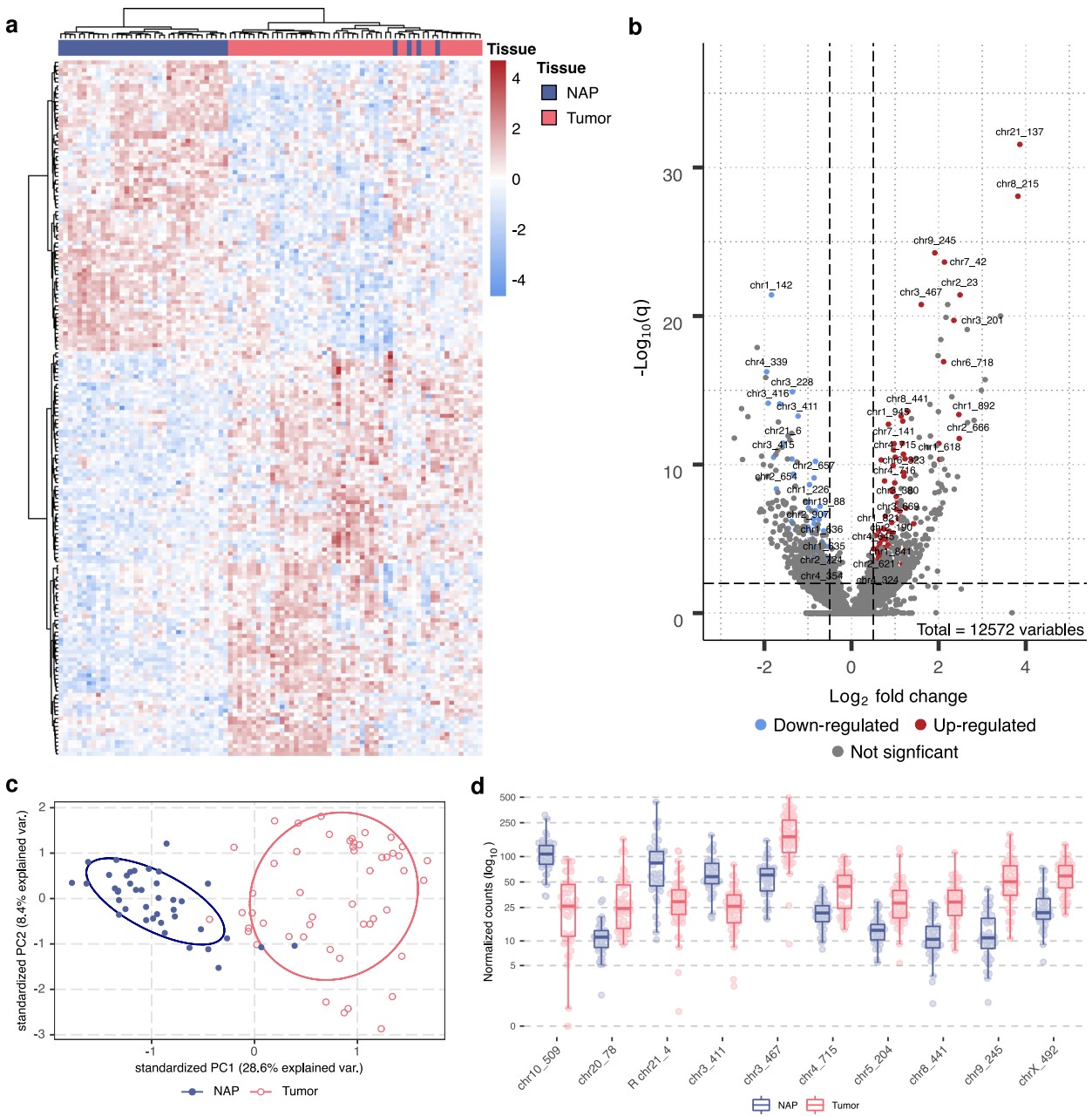

**Fig. 5 Prostate cancer-specific expression of the hCRISPRs in the NGS-ProToCol dataset. a** Unsupervised clustering (Euclidean distances; Ward.D2) of malignant (orange; top-bar) and healthy (blue; top-bar) prostate tissues using normalized read counts (VST) as Z-scores over all 177 statistically significant hCRISPRs. ($|\log_2$ fold-change$| \geq 0.5$, average read-count over all samples $\geq 10$, and $q \leq 0.05$). Negative Z-scores are highlighted in green whilst positive Z-scores are highlighted in red. **b** Volcano-plot depicting the $\log_2$ fold-change (x-axis) and adjusted p-value (q) (y-axis; in $-\log_{10}$ scale) of all 12,572 hCRISPRs. The hCRISPRs which were found to be differentially upregulated ($\log_2$ fold-change $\leq 0.5$, average read-count over all samples $\geq 10$ and $q \leq 0.05$) and downregulated ($|\log_2$ fold-change$| \geq 0.5$, average read-count over all samples $\geq 10$ and $q \leq 0.05$) in cancer are shown by red and blue dots, respectively. The hCRISPR identifier (#Order) is shown for the top 25 most substantial (based on adjusted p) upregulated and downregulated hCRISPRs. **c** Overview of the stratification of the NGS-ProToCol cohort using principal component analysis (PCA) on the 177 differentially expressed hCRISPRs (using VST-normalized read counts) with the first two principal components (PC1 and PC2). Malignant prostate tissues are depicted by salmon points whilst the normal adjacent to tumor prostate tissues are depicted by blue points. **d** Boxplots representing the normalized expression (VST-transformed read-count) of the top ten hCRISPRs with an absolute $\log_2$ fold-change $\geq 1$ and average read counts $\geq 20$, ordered on descending q-value; median, Q1 and Q3 are highlighted with a bold black line and error bars, respectively. Malignant prostate tissues are depicted by salmon points whilst the normal adjacent to tumor prostate tissues are depicted by blue points. Normalized expression (VST-transformed read-count; y-axis) is shown in $\log_{10}$ scale.

recombination of immunoglobulin genes and regulation of chromatin architecture[59,60]. This could suggest that one of the functions that some of the hCRISPR fulfill, is a role in maintenance of chromosomal structures, a process disrupted in cancer[61], or maybe in a more centromere-like-related function, as suggested for the CRISPR arrays in archaea[3,8]. In laymen terms, fulfilling roles as anchors to separate chromosomes from each other.

In this work, we identified 12,572 hCRISPRS and discovered that a majority of these elements are expressed in a tissue-specific manner. Of main importance is our discovery that their

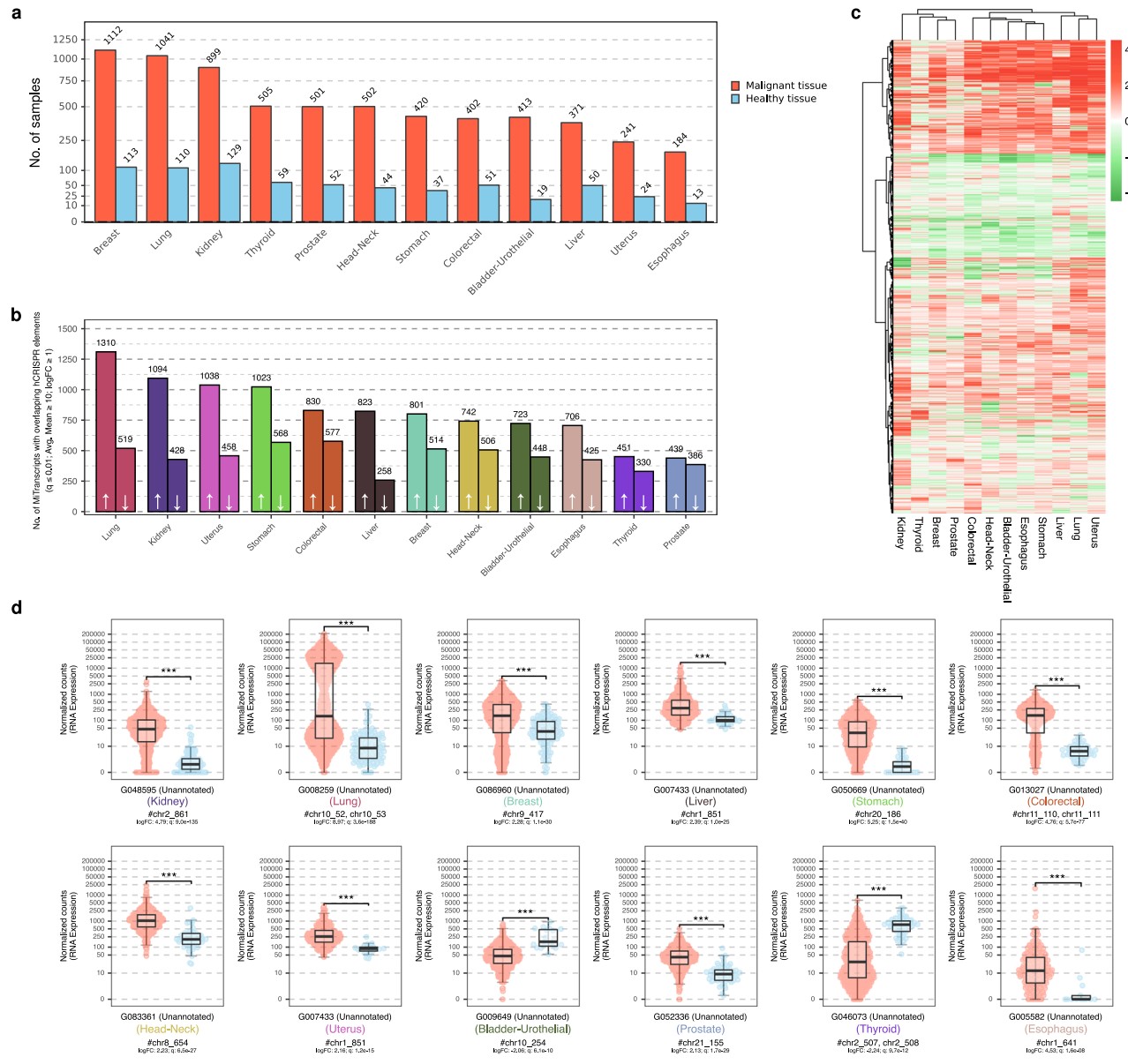

**Fig. 6 Cancer-specific expression of MiTranscripts overlapping hCRISPR within the MiTranscriptome cohort. a** Overview of tissue distribution per tumor subtype within the MiTranscriptome dataset which consisted of at least 10 healthy and 10 malignant samples. **b** Number of differential MiTranscriptome transcripts between healthy and malignant tissue per malignant tissue ($|\log_2$ fold-change$| \geq 1$, an average read-count of at least 10 over all samples in the respective tissue and $q \leq 0.01$). Number of transcripts are shown per direction (up (left) / and downregulated (right)) as separate bars. **c** Unsupervised clustering (Euclidean clustering; Ward.D2) of all distinct differential Mitranscriptome transcripts ($n = 4622$) using their normalized expression (VST). Negative Z-scores are highlighted in green whilst positive Z-scores are highlighted in red. **d** Boxplots representing the normalized expression (VST-transformed read-count) of a representative unannotated Mitranscriptome transcript overlapping with a single hCRISPR per distinct malignant tissue; median, Q1 and Q3 are highlighted with a bold black line and error bars, respectively. Normalized expression (VST-transformed read-count; y-axis) is shown in $\log_{10}$ scale.

expression enables the differentiation between malignant and normal adjacent to tumor tissue, and thus provides an opportunity to differentiate patients from healthy individuals. This opens the door to explore the usage of hCRISPRs in diagnostics and prognostics to detect a wide variety of diseases, including cancer. In prostate cancer, we already made advantage of this discovery, as we identified 115 hCRISPRs (nine without any overlapping gene) that are candidate biomarkers as validated in two independent prostate cancer related datasets (Mitranscriptome and NGS-ProToCol)[40,42–44,62]. For two of these hCRISPRs (chr9_209 and

chr19_106), we further performed and established the ability to distinguish malignant from normal adjacent to tumor prostate tissue by RT-qPCR, confirming whole-transcriptome results of the same tissues reported earlier[38,39]. At that time, we did not know that these transcripts belonged to an underlying hCRISPR. Moreover, we anticipate that the SHERLOCK[47] and DETECTR[48] technologies based on the expression of the hCRISPRs will enhance the development of affordable point-of-care tests for preventive screening measures in cancer and other human diseases.

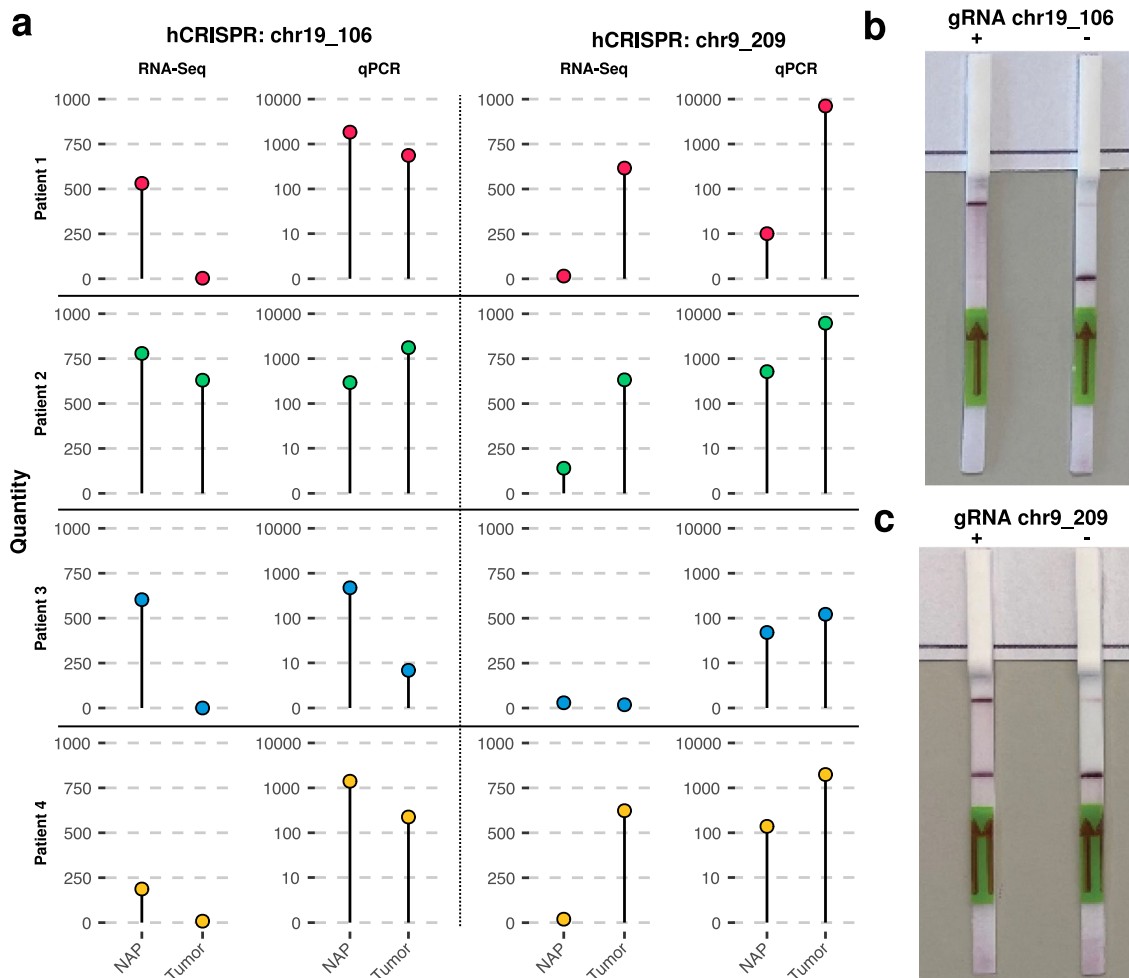

**Fig. 7 RT-qPCR validation of hCRISPR expression in normal prostate tissue and tumor samples from four prostate cancer patients and detection by SHERLOCK and DETECTR. a** Expression of hCRISPR chr19_106 and chr9_209. Left panels show the normalized number of sequencing reads mapping to hCRISPR chr19_106 and chr9_209 in the NGS-ProToCoL RNA-Seq dataset. Right panels show the normalized transcript copy number of hCRISPR chr19_106 and chr9_209 measured by RT-qPCR in the same samples. NAP is normal adjacent to tumor prostate tissue. Tumor, prostate tumor tissue. **b** Photograph of the DETECTR technology identifying the presence of hCRISPR chr19_106. (+) indicates a positive and (−) indicates a negative detection result. **c** Similar to **b** but for the presence of hCRISPR chr9_209.

## Methods

**Identification of CRISPRs in the human reference genome.** The genomic sequence of all standard autosomal and sex chromosomes (chr1 to chrY) of the human reference genome (GRCh38.p13; GCF_000001405.39) was analyzed using the stand-alone CRISPR identification software tool CRISPRCasFinder[20] (v4.2.19; CasFinder v2.0.2) with default settings, except for the following parameters: -minSP 21 (min. size of spacer) -maxSP 72 (max. size of spacer) -fast, searching for additional *cas* genes was included. Identified hCRISPRs were also lifted over to GRCh37 coordinates using the rtracklayer package (v1.46.0) with UCSC Chain Files (hg38ToHg19.over.chain) for compatibility with MiTranscriptome. To compare the CRISPRCasFinder software[20] findings, we also run the human genome (GRCh38) in the software tools CRISPRDetect[63] and CRISPRCasTyper[64] in default settings except for CRISPRDetect, in which we set the minimum number of repeats at two and the CRISPR likelihood score at zero. CRISPRMap[65] was used to further analyze the hCRISPR consensus repeats retrieved from the CRISPRCasFinder, CRISPRDetect and CRISPRCasTyper software tools to identify Cas-related endo-nuclease motifs, RNA repeat structure and sequence families and CRISPR-Cas superclasses. CRISPRloci[66] was used to determine whether the consensus repeats harbored identity with prokaryotic consensus repeats as deposited in the database of this software tool. Both CRISPRMap and CRISPRloci were run at default settings.

For associating the hCRISPRs with other organisms, a BLAST search was performed with all detected hCRISPRs against the non-redundant nucleotide database in NCBI. For each BLAST hit with a hCRISPR, a taxonomical unit was assigned based on NCBI taxonomy. Visualization of the hCRISPRs and association with other organisms was performed in R. hCRISPRs that correlated to mammals were clustered based on their sequence using CD-HIT[67] and were further visualized in a clustering network using ggnetwork in R.

For identification of *cas* gene signatures we retrieved the flanking regions 20,000 bp upstream or downstream of the hCRISPR. We then applied the 3D uniport BLAST tool (https://www.uniprot.org/blast/) combined with the myhits motif scan (https://myhits.sib.swiss/cgi-bin/PFSCAN), the latter used successfully earlier to detect the nuclear localization signal in *Campylobacter jejuni* Cas9[68]. The same length of the flanking regions was used to search for the RVT_1 (PF00078) element by using the *cas* gene repository published earlier[69,70] and a local BLASTx analyses (e-value: 1e-06). The results were further screened for >50% coverage of the subject (database) and >40% identity and the query hit with the maximum bit-score was considered as the final result.

**Determining the overlap of hCRISPRs with ENCODE datasets.** The hCRISPRs were overlapped, based on a minimum overlap of five base pairs, with genes derived from GENCODE[71] (GRCh38, v33) containing transcripts with transcript support levels (TSL) of 1 and/or 2 ($n = 1,417,004$). Furthermore, the hCRISPRs (with liftOver coordinates to GRCh37) were overlapped, based on a minimum of five base pairs, with repeat regions ($n = 5,467,457$) derived from RepeatMasker (GRCh37, Feb 2009)[72].

Genomic foci of TFBS ($n = 4,380,444$) from 161 factors, DNase I Hypersensitivity Clusters ($n = 1,867,665$) derived from 125 cell types and CpG islands ($n = 28,691$) were downloaded from the ENCODE project through the UCSC Genome Browser (GRCh37) repository (wgEncodeRegTfbsClusteredV3wgEncode RegDnase, ClusteredV3, and CpGIslandExt tracks, respectively) and the hCRISPRs were overlapped, based on a minimum of five base pairs. We then utilized regioneR[73] to determine if the hCRISPRs were found to be enriched or depleted within a set of genomic regions. We performed regioneR using default settings using 5000 iterations per analysis and performed this for the set of CpG islands, DNase I clusters, POLR2A-

sites, and CTCF-sites as these held the highest overlap with the CRISPRs identified in the human genome.

To detect surrounding polyadenylation signals the genomic sequences of the hCRISPRs and surrounding regions (100 bp up-/downstream) were obtained from the human reference genome (GRCh38.p13) and processed by POLYAR to predict putative poly(A) sites including cleavage/polyadenylation sites within these regions[31].

The DASHR 2.0 ($n = 191,966$)[74], piRBase v2.0 ($n = 1,077,308$)[33] databases and RNAcentral ($n = 586,927$)[34] databases (filtered on autosomal and sex chromosomes) containing genomic annotation of various noncoding classes, such as PIWI-interacting RNAs, microRNAs, small nuclear RNAs, transfer RNAs, small nucleolar RNAs, ribosomal RNAs, and sncRNAs, were overlapped with hCRISPRs based on a minimum of five base pairs. In addition, Erasmus Medical Center Prostate Cancer Associated Transcripts, better known as EPCATs, as published in Böttcher et al.[43], are noncoding RNAs usable as diagnostic and prognostic markers in prostate cancer ($n = 437$) and were also overlapped with the hCRISPRs based on a minimum of five base pairs.

**Characterizing the expression of the hCRISPRs.** To determine the expression of the hCRISPRs within various human cell lines and prostate cancer, we utilized the small-RNA-sequencing datasets, obtained from Encyclopedia of DNA Elements better known as ENCODE[75], of the human cell lines K562, HeLa (two replicates), Hep-G2 and HUV-EC-C (two replicates) and a small-RNA-sequencing dataset capturing malignant an normal adjacent tumor to prostate tissue (GSE80400)[38,39]. The exact condition of cell line culturing and RNA-sequencing mapping of the human cell lines can be found under the accession numbers; (1) GSM605630 for the K562 cell line; (2) GSM897079 for the HELA S3 cell line; (3) GSM897084 for the Hep-G2 cell line, and (4) GSM897075 for the HUV-EC-C cell line. Raw sequence reads were downloaded as-is, and any remaining sequence adapters were trimmed using Cutadapt (v2.8) at default settings with a list of common (small-) RNA-sequencing adapters and discarding reads with remaining sequence lengths below 15 base pairs.

Bowtie indexes were generated based on reads from the trimmed small-RNA reads from eukaryotic cell lines and prostate cancer datasets. Against these reads were mapped the hCRISPRs and surrounding regions (100 bp up-/downstream), as obtained from the human reference genome (GRCh38.p13) using bowtie[76] with parameters -N 0 -L 20 -i 'S,1,0.50' --n-ceil 'C,0' --dpad 15 --gbar 4 --no1mm-upfront --end-to-end --score-min 'C,0' -D 20 -R 3. The SAM format output was converted to BAM format using SAMtools[77].

RNA-sequencing data for 185 tissues and cell lines were downloaded from DASHR (http://dashr2.lisanwanglab.org/download.php)[32]. The BED files containing genomic annotation of RAW expression profiles for noncoding RNAs across tissues and cell lines in read counts per mature sncRNA or reads per million were overlapped with the hCRISPRs based on a minimum of five base pairs. In addition, we made advantage of the DASHR v2.0 UCSC Genome browser track hub [GRCh38/hg38] in which all the processed sequencing data is effectively visualized in the UCSC genome browser. In combination with uploading the genomic positions of the hCRISPRs in a BED file format as a custom user track, we could check for DASHR 2.0 sncRNAs that originated from the hCRISPRs.

Whole-transcriptome data of 16 distinct health tissues from the Illumina human BodyMap 2.0 dataset[36,37], aligned to the human reference genome (GRCh37), was downloaded (GSE30611) and used as-is to determine expression status of the hCRISPRs. This cohort consisted of distinct healthy tissues from the human brain, colon, heart, kidney, lung, liver, thyroid, white blood cells, skeletal muscle, adrenal gland, lymph node, ovary, testes, adipose, breast, and prostate.

The number of overlapping uniquely mapped/primary reads per hCRISPR per input BAM file was counted using the Rsamtools package (v 1.34.1)[78]. As two out of four cell lines (K562 and HeLa S3) are from female donors, hCRISPRs on the chromosome Y were ignored for overlap in the cell line analysis. Reads per hCRISPR were converted to transcript per millions (TPM) with the following formula:

$$\text{TPM}_i = \frac{r_i}{w_i} \cdot \left( \frac{1}{\sum_j \frac{r_j}{w_j}} \right) \cdot 1E6 \qquad (1)$$

where $r$ stands for total number of unique-mapped reads overlapping the hCRISPR and $w$ the length of the hCRISPR in kilobases. hCRISPRs with ≥50 TPM in one of the 16 BodyMap 2.0 tissues or cell lines were considered to be expressed in that particular tissue and/or cell line.

**Validating the hCRISPR biomarker potential in localized prostate cancer using the NGS-ProToCol cohort.** Prior to alignment, from the NGS-ProToCol dataset (EGAS00001002816), sequence adapters (TruSeq3) were trimmed using Trimmomatic[79] (v0.38) and paired-end reads were subsequently aligned to the human reference (GRCh38.p12) using STAR[80] (v2.7.0a) with genomic annotations from GENCODE[81] release 29. Generation of alignment quality metrics (flagstat) and duplicate reads marking was performed by Sambamba[82] (v0.6.7). FeatureCounts[83] (v1.6.0) was used to generate raw read-count tables for each hCRISPR; only primary (uniquely mapped) reads were counted per hCRISPR using paired-end modus. Normalization and differential analysis on the 12,572

hCRISPRs was performed using DESeq2[84] (v1.24.0) between malignant and normal adjacent to tumor prostate tissue. To correct for multiple hypothesis testing after DESeq2 analysis, we employed independent hypothesis weighting (IHW; v1.12.0)[85]. Fold-changes ($\log_2$) were shrunken using their respective coefficient using apeglm (v1.6.0)[86]. We used the following criteria to determine statistically significant differentially expressed hCRISPRs: |$\log_2$ fold-change | ≥ 0.5, adjusted $p \leq 0.05$ and an average read-count over all samples ≥10.

**Validating the hCRISPR expression in malignant and normal adjacent to tumor prostate tissue using the RT-qPCR technology.** Eight samples of matched malignant and normal adjacent to tumor prostate tissue from four patients with prostate cancer were obtained from the Erasmus MC tissue bank. Use of the samples for research purposes was approved by the Erasmus MC Medical Ethics Committee according to the Medical Research Involving Human Subject Act (MEC-2004-261; MEC-2010-176) and patients were notified via an informed consent and asked for permission. Patients were selected based on the availability of RNA Seq data, and evaluation of hCRISPR expression levels (RPM) in each sample. From each sample total RNA was isolated. cDNA synthesis was performed using MiRCURY LNA RT kit (QIAGEN, Venlo, Netherlands, Cat. No. / ID: 339340) and in each RT-reaction 200 ng total RNA was used. RNA spike in template was added to each of the RT mixes according to manufacturer instructions. Quantitative real-time reactions were performed with custom designed primer assays (QIAGEN, Venlo, Netherlands Ct. No: 339317). Template sequence and Assay IDs are chr19_106 (5'-CTGACTAATACAGATTTTGGCACCAG-3') Gen-eGlobe Design ID YCP1372825 and chr9_209 (5'-AGAGATATTCTTA-GAATCTTTCATTATGGTACTCATAT-3') GeneGlobe Design ID YCP-1372891. Real-time PCR reactions were performed with SensiMix™ SYBR® Low-ROX kit (Meridian bioscience, Boxtel, Netherlands) on an ABI 7500 Fast Real time PCR system (ThermoFisher, Breda, Netherlands). Four microliters of 60 times diluted cDNA was used for triplicate real-time PCR reactions. Cycling conditions were: Enzyme activation for 10 min at 95 °C, followed by 45 cycles of 15 sec at 95 °C, 30 sec at 55 °C, 15 sec at 72 °C. For the generation of standard curves for absolute quantification, synthetic 5'-phosphorylated RNA oligos mimicking the detected hCRISPRs were purchased (IDT, Breda, Netherlands). The oligo templates were mixed in an equimolar ratio and 10-fold serial dilutions were prepared over five orders of magnitude from 10 pM (~10 mln. copies) to 10 fM (~1000 copies). Real-time PCR was performed in high sensitivity AmpliStar-II 96-well plates (Westburg, Leusden, Netherlands, Cat. No: 1900WS). To detect chr9_209 with the SHER-LOCK technology we used the Cas13a protein (MCLAB, San Francisco, USA). First, the following reaction mixtures were prepared. To generate a working stock solution of the sgRNA (5'-GAUUUAGACUACCCCAAAAACGAAGGGGA-CUAAAACUAAUGAAAGAAUUCUAAGAAU-3') (Synthego, Redwood city, USA) to detect the chr9_209 transcript (5'-AGAGATATTCTTAGAATCTTTCATTA TGGTACTCATAT-3'), we added to the 5 nmol stock 20 µl of nuclease-free 1× TE buffer (Tris-Edta, pH8.0) to get a final concentration of 250 µM (250 pmol/µl). To make a 10 µM working stock, we add 5 µl of the 250 µM sgRNA oligo to 120 µl of nuclease-free water to get a total volume of 125 µl of a 10 µM sgRNA working stock solution. Then a 1× NEBuffer 2.1 solution was prepared by mixing 100 µl of 10× stock NEBuffer 2.1 with 900 µl of nuclease-free water, to this solution 20 mg PEG 6000 was added. The reporter was diluted to get a 100 µl 10 µM solution by mixing 10 µl of 100 µM oligo stock with 90 µl of nuclease-free 1× TE buffer (Tris-EDTA) buffer. Then a cleavage reaction mixture was prepared by combining 2 µl cleavage buffer (400 mM Tris pH 7.4), with 12.8 µl UltraPure water, 0.5 µl of Cas13a (1 mg/ml, MCLAB), 1 µl of the gRNA (10 µM), 1 µl MgCl2 solution (120 mM), and 0.7 µl, of the Cas13 reporter (10 µM) (Lateral-Flow-Reporter: 5'-/6-FAM/rUrUrUrUrUrUrUrUrUrUrUrUrUrU/Bio/-3') (IDT, Breda, Netherlands)). Then 2 µl of the chr9_209 synthetic transcript (1 µM) was added to the cleavage mix and incubated at 37 °C for 35 .min Hereafter 80 µl of 1× NEBuffer 2.1 with 2% PEG was added to each 20 µl reaction mixture and mixed thoroughly. We then placed the diluted reaction in a tube rack at room temperature, in which a HybriDetect Dipstick was added. The lateral-flow strip was allowed to run for two minutes at room temperature, where after the result was rated and picture captured at three minutes.

To detect chr19_106 with the DETECTR technology we used the EnGen® Lba Cas12a (Cpf1) protein (M0653T) (New England Biolabs, Ipswich, USA). First, the following reaction mixtures were prepared. To generate a working stock solution of the sgRNA (5'-UAAUUUCUACUCUUGUAGAUUGGUGCCAAAAUCUGUAU UA-3') (Synthego, Redwood city, USA) to detect the chr19_106 transcript (5'-CT GACTAATACTAATACAGATTTTGGCACCAG-3'), we added to the 5 nmol stock 20 µl of nuclease-free 1x TE buffer (Tris-EDTA, pH8.0) to get a final concentration of 250 µM (250 pmol/µl). To make a 10 µM working stock, we added 5 µl of the 250 µM sgRNA oligo to 120 µl of nuclease-free water to get a total volume of 125 µl of a 10 µM sgRNA working stock solution. Then a 1× NEBuffer 2.1 solution was prepared by mixing 100 µl of 10x stock NEBuffer 2.1 with 900 µl of nuclease-free water. To this solution 20 mg PEG 6000 was added. The reporter was diluted to get a 100 µl 10 µM solution by mixing 10 µl of 100 µM oligo stock with 90 µl of nuclease-free 1× TE buffer (Tris-EDTA, pH8.0). To detect the transcript of chr19_106 a LbaCas12a working stock of 10 µM was generated by adding 1 µl of 100 µM LbaCas12a stock to 9 µl nuclease-free water. Then a cleavage mix was prepared with per sample 13.3 µl nuclease-free water, 2.0 µl 10× NEBuffer 2.1,

0.8 µL of 10 µM LbCas12a, 1.0 µl of 10 µM gRNA and 0.9 µl of the Cas12 reporter (10 µM) (Lateral-Flow-Reporter: 5'-/56-FAM/TTATTATT/3Bio/-3') (IDT)). Two microliter of the DNA chr19_106 transcript was then added to this cleavage mixture and incubated at 37 °C for 35 min. Hereafter 80 µl of 1× NEBuffer 2.1 with 2% PEG was added to each 20 µl reaction and mixed thoroughly. A HybriDetect Dipstick was then placed into each reaction tube and allowed to run for two minutes at room temperature into the lateral-flow strip, where after the result was rated and picture captured at three minutes.

**Validating the hCRISPR biomarker potential in localized cancer using a Pan-cancer MiTranscriptome RNA-sequencing dataset.** To identify biomarker potential in localized cancer settings, we investigated the MiTranscriptome dataset, which is a large-scale ab initio transcriptome meta-assembly from 10,225 RNA-Seq libraries derived from 36 distinct malignant tissues[44]. Raw reads per gene, gene, and transcript annotation and sample metadata were combined and converted using Summarized Experiment R package to ease downstream analysis in R. The MiTranscriptome dataset was filtered to only contain tumor subtypes in which ≥10 normal and ≥10 malignant tissues were available on which differential analysis could be performed, this heuristic filtering left 12 out of 70 distinct tumor subtypes for downstream analysis (lung, kidney, colorectal, stomach, breast, head/neck, uterus, liver, bladder/urothelial, thyroid, prostate, and esophagus).

Differential gene analysis between healthy and malignant tissue, per tissue, was performed using DESeq2 (v1.22.2)[44] with the Wald test and Benjamini–Hochberg correction on all MiTranscriptome transcripts overlapping hCRISPRs with a minimum of five base pairs ($n = 7750$). Significant results of the differential analysis were obtained with the following criteria: $|logFC| \geq 1$ and adjusted $p$-value (BH) $\leq 0.01$ and an average read-count, over all samples, of at least 10. In total 3850 distinct MiTranscriptome transcripts overlapping hCRISPRs were observed over all tissues.

**Quantifying the expression of the hCRISPRs during human microbial infection and other diseases.** To test any potential roles of the proposed hCRISPRs ($n = 12,572$) in human diseases, we analyzed the expression of the Human Genome U133 Plus 2.0 Array noncoding probes in various diseases, which overlapped or exactly matched these hCRISPRs (Supplementary Data 1, 10 and 11). In (Supplementary Fig. 4) additional examples are given that include GDS4966—Active TB[87], GDS3615—Asthma[88], GDS3902—Chronic B-lymphocytic leukemia[89], GDS3298—*Francisella* infection[90], GDS4882—Hepatocellular carcinoma[91], GDS5017—Newly diagnosed and chronic pediatric immune thrombocytopenia[92], GDS1439—Prostate cancer[93], GDS2697—Teratoospermia[94] to visualize the differences in RNA expression value between the different conditions tested. The RNA expression datasets of the visualized examples are presented in (Supplementary Data 10). Per dataset, we quantified the expression values according to following method; for our analyses the probe identification numbers, expression values and sample IDs were retrieved from the dataset and uploaded into Galaxy[95] as a BED file dataset, which was adapted for the human assembly GRCh37. The hCRISPR-overlapping or exactly matching noncoding U133 plus 2.0 probes (Supplementary Data 1) were uploaded as a BED file dataset and adapted for the human assembly GRCh37 and merged with the expression values using the "join two datasets" to generate the intersection of the hCRISPR-overlapping U133 probes and U133 RNA expression values. Using the metadata, as provided by the original authors, we grouped samples into the various dataset-specific conditions. Example given, untreated blood monocytes and blood monocytes infected with *Francisella tularensis* subsp. Novicida or *Francisella tularensis* subsp. Tularensis in the GDS3298 dataset[90]. The hCRISPR-overlapping or exactly matching U133 plus 2 probes and their RNA expression values were than imported into MORPHEUS (at https://software.broadinstitute.org/morpheus/)[96] to generate a mean-centered heatmap showing expression differences, which was used to visually determine potential biomarkers between the dataset-specific conditions. Tukey's Multiple Comparison Test or Wilcoxon signed-rank test (two-sided) was used to test statistical significance between each pairwise comparison for the hCRISPR-overlapping probes of interest.

**Statistics and reproducibility.** Visualization, quantification, and statistical analysis has been performed in the R statistical platform language (v3.6.0 and v4.1.0). Unless stated otherwise, statistical significance was calculated using a Mann–Whitney test with Benjamini–Hochberg (BH) correction. When shown, adjusted $p$-values were depicted as followed: adjusted $p < 0.05$ as *, adjusted $p < 0.01$ as **, and adjusted $p < 0.0001$ as ***.

**Reporting summary.** Further information on research design is available in the Nature Research Reporting Summary linked to this article.

## Data availability
The publicly available datasets can be requested from their respective repositories under the accession-number as described within the methods, including GSE80400, GSE30611, GSE80400, GSM605630, GSM897079, GSM897084, and GSM897075 from NCBI GEO. The MiTranscriptome dataset can be requested from the original authors upon reasonable request. Custom data relating to the qPCR experiments and analysis have been deposited within the Supplementary Information file. The underlying data for Figs. 2–4, 5d, 6 has been deposited within Supplementary Data 1 and Supplementary Data 5. The original data underlying Fig. 7 has been deposited within Supplementary Data 12.

## Code availability
All custom code used in the analysis has been deposited upon Zenodo: https://doi.org/10.5281/zenodo.6122533

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

## Acknowledgements

C.S. is a graduate student at Erasmus Postgraduate School of Molecular Medicine and is partially supported by I&I Fund (Erasmus Vrienden Fonds). The bioinformatics team of P.J.v.d.S. receives supporting funding from H2020 projects ImmuneAID and Mood-stratification for Data analysis infrastructure. Moreover, national funding has been obtained from the ZonMW project Genomes First for genomics infrastructure. The team of R.L. receives supporting funding from H2020 project STAMINA (ID: 883441), the Chiron Foundation and by PPP Allowance made available by Health~Holland, Top Sector Life Sciences & Health, to stimulate public-private partnerships LSH-TKI foundation grant LSHM18006. Prostate RNA-seq data were generated within the framework of the CTMM research program, project NGS-ProToCol (grant 03 O-402). We acknowledge Rene Akre for critical reading and editing the manuscript. Finally, we would like to pay our deepest respect to Prof. Johan W. Mouton, who passed away on the 9 July 2019 due to complications caused by prostate cancer (https://www.thelancet.com/journals/laninf/article/PIIS1473-3099(19)30441-4/fulltext). Prof. Mouton was supervisor of author and PhD student Chinmoy Saha, thanks to his support, guidance, motivation and scientific advice this work might form the basis to improve prostate cancer diagnostics and prognostics.

## Author contributions

R.L., C.S., J.v.R., G.J., R.W.W.B., W.I.J., and P.J.v.d.S. designed the general research questions to detect and validate the hCRISPRs in transcription and diagnostics. R.L., C.S., and J.v.R. wrote the manuscript, which all authors critically reviewed. J.v.R., N.S., R.W.W.B., Y.H., Y.Ha., S.M.A.S., A.S., A.M.T., and R.L. performed the computational analyses. Y.S.N. and A.M.C. analyzed and organized and provided the MiTranscriptome datasets. W.I.J. coordinated the sequencing of samples as addressed in earlier related work. E.S.M., M.A.K., Y.H., and R.J.A.F. sequenced, analyzed, and provided the NGS-ProToCol-related datasets. E.S.M. designed and performed the RT-qPCR analyses. B.J. performed the genome sequencing of the NGS-ProToCol samples. S.V. tested, optimized, and run the SHERLOCK and DETECTR technologies. All authors have read, edited, and agreed to the published version of the manuscript.

## Competing interests

The authors R.L., P.J.v.d.S., and C.S. declare the following competing interests, as they are included as inventors on CRISPR-Cas-related patents. The remaining authors declare no competing interests.
