## [Peer Review File · Communications Biology]

Reviewers' comments:

Reviewer #1 (Remarks to the Author):

The authors have subjected the DNA sequences of the human genome to a search for the presence of repeated sequences by computational methods. The repeat sequences searched for had the structure of the SRSR sequences described in 2000 by Mojica et al in Molecular Microbiology. In addition to the identification of the SRSR structures in the DNA they searched several expression databases for the presence of RNA transcripts of these repeat sequences and linked these to several diseases.

Major comments:

The SRSR repeats described by Mojica et al in 2000 are nowadays known as CRISPR. It was stated in the publication by Jansen et al (2002) 43 1565-1575 in Molecular Microbiology that the SRSR repeats and the CRISPR are identical and that in future publication the CRISPR acronym will be used for these peculiar repeats. Mojica and many others refer to these repeats as CRISPR and therefore it is surprising to stumble upon the SRSR acronym in the manuscripts. Moreover, the SRSR sequences are identified with the CRISPR-CasFinder software and it is strange to read that this package is described as an SRSR identification tool (lines 308-309).

1- It is recommended that the authors rewrite the manuscript using the CRISPR acronym and discuss their findings in the light of the wealth of knowledge that is available about the CRISPR and its accompanying cas genes. The identification of the human repeat sequences as SRSR is not convincing as the authors do not show any of the characteristic spacer sequences that are so well known in the CRISPRs (and hence SRSRs). It can be expected that the human repeats are better described as imperfect repeats rather than SRSRs.

2- Actually, the authors only show the presence of repeat sequences in the human genome, a well known feature and they refer to reports and databases in which these repeat sequences are transcribed in certain cell lines and tumor tissues. The authors suggestion that these SRSR's can be used as biomarkers might be useful. It is recommended to rewrite the manuscript and focus on this subject.

Reviewer #2 (Remarks to the Author):

Here, the authors describe the identification in humans of a class of repeat previously thought to be limited to prokaryotes and archaea. I focused primarily on the evolutionary history and genomic distribution of SRSRs in humans as part of my review and found that I would appreciate a better exploration into the evolutionary history of such elements in the human genome. In particular, how do the authors suggest that these elements arose in the human genome and, in their eyes, gained a function? I have several comments that I believe the authors should address:

Major Concerns:

- The authors present no reasoning for why SRSRs were previously thought to not exist in Eukaryotic genomes, both in the context of the Introduction and in the context of their own work as part of the Results/Discussion. Further to this, are SRSRs identified by this study conserved among closely related Eukaryotes (i.e. Great Apes) or more distantly related species (mouse comes to mind)? When ascribing function to something as evolutionarily ancient as SRSRs, it is unlikely that this functionality arose completely de novo in the human species.

- What is the sequence homology of the repeat sequences of identified SRSRs? Do you identify families of SRSRs in humans?

- Overall, I find the manuscript lacking for statistics when terms like "enrichment" and "substantial" are used. Many of the proportions presented are out of context without a null expectation (i.e. 41.6% with polyA recognition sites seems high, but the most common poly(A) sequence homology is AATAAA and as such is likely to be fairly common in the human genome). In particular, it is unclear to me if SRSRs overlap with particular consequences/genomic

features/binding sites more than expected by chance. It would be helpful to update Figures 2/3 with such an analysis. Specifically, the authors state "we noticed an enrichment for RNA Polymerase II Subunit A (POLR2A) and the CCCTC-Binding factor (CTCF)" but do not provide any sort of statistic to show such an enrichment. Further to this, I have no context for the number of SRSR elements that should be found to be expressed in any of the GEO databases cited in the study.

- It is unclear to me whether SRSRs are expressed independently of the transcripts within which they are contained. Is the differential expression you observe in cancer genomes simply due to expression of such transcripts? The author's statement about EPCATs suggests this to be true and as such what is the actual diagnostic potential of SRSRs beyond EPCATs?

- Do you have any molecular evidence to show that SRSRs are expressed as distinct molecules (e.g. Northern Blotting/RT-PCR)?

- While SRSRs may overlap with peaks for TFBS¹, does the actual spacer/repeat sequence harbour any homology to any known transcription factor?

- The statement in the discussion "Of main importance is our discovery that their expression enables 256 the differentiation between healthy individuals and diseased patients" does not appear to be supported by anything beyond circumstantial evidence. Can the authors show that a predictive model using a test/training regimen is predictive and that it is more predictive than the literature cited in the manuscript (i.e. Böttcher, R. et al.)?

Minor Comments:

- This is more of a scientific writing style choice, but I find the use of superlative words in the manuscript like "mysterious", "remarkable", and "peculiar" as scientifically imprecise (particularly in the Introduction). For example: what does a word such as "mysterious" actually mean in the context of "researchers identified a mysterious class of repeats" (L. 52) ? Does it mean unexpected, surprising, dissimilar to other repeat classes, etc.? I would prefer the authors be more specific in their writing.

- I am unsure why the section "Diagnostic potential of RNA transcripts..." begins with a large summary of the author's previous work. A one sentence summary with a citation would suffice.

- Generally, grammar needs editing/correcting. Quick examples include:

o L.71: "potentially, even via a more" ♦ "potentially via a more"

o L. 165-6: "the SRSR elements made use investigate their 166 expression activity in more detail." (I'm not sure what this sentence is saying)

- sncRNAs should be defined when "small non-coding RNAs" is first used on L.78. It is currently defined on L. 149. The acronym is also not used in several places when it has been defined already (e.g. L.145-6, L. 199, etc.)

- The discussion of CRISPR in the discussion seems out of context with the rest of manuscript as it is never mentioned anywhere else. It might be worth giving context in the Introduction so this is less jarring.

- I feel the use of "new" in the title of the manuscript is not particularly accurate. They may be "new" to the human genome but are not "new" in an evolutionary or scientific sense. Additionally, what does "beyond" mean in this context?

Author comment:

First of all, we would like to thank both reviewers for their feedback and constructive suggestions that certainly improved our manuscript significantly.

Reviewer #1 (Remarks to the Author):

The authors have subjected the DNA sequences of the human genome to a search for the presence of repeated sequences by computational methods. The repeat sequences searched for had the structure of the SRSR sequences described in 2000 by Mojica et al in Molecular Microbiology. In addition to the identification of the SRSR structures in the DNA they searched several expression databases for the presence of RNA transcripts of these repeat sequences and linked these to several diseases.

Major comments:

The SRSR repeats described by Mojica et al in 2000 are nowadays known as CRISPR. It was stated in the publication by Jansen et al (2002) 43 1565-1575 in Molecular Microbiology that the SRSR repeats and the CRISPR are identical and that in future publication the CRISPR acronym will be used for these peculiar repeats. Mojica and many others refer to these repeats as CRISPR and therefore it is surprising to stumble upon the SRSR acronym in the manuscript. Moreover, the SRSR sequences are identified with the CRISPRFinder software and it is strange to read that this package is described as an SRSR identification tool (lines 308-309).

It is recommended that the authors rewrite the manuscript using the CRISPR acronym and discuss their findings in the light of the wealth of knowledge that is available about the CRISPR and its accompanying cas genes. The identification of the human repeat sequences as SRSR is not convincing as the authors do not show any of the characteristic spacer sequences that are so well known in the CRISPRs (and hence SRSRs). It can be expected that the human repeats are better described as imperfect repeats rather than SRSRs.

Response:

Based on the reviewer suggestions, we decided to adapt the SRSR acronym used in the previous version of the manuscript to CRISPR in the revised version. We do caution this finding by clarifying that we do not provide further proof for a biological function such as prokaryotic defence mechanisms (**see Supplementary Information**). To better address that the CRISPRs identified in the human genome contain the characteristic CRISPR signature rather than being some form of imperfect repeat, we have added the spacer sequences for each CRISPR in the **Supplementary Data 1** file. As depicted in **Supplementary Fig. 1** two example CRISPRs are both shown to harbor the characteristic palindromic repeat and spacer sequences, which hallmarks the known CRISPR-signature.

In addition, we also employed two alternative CRISPR-detection algorithms (CRISPRDetect and CRISPRCasTyper), which detected subsets of these CRISPRs that were identified by the CRISPRCasFinder. CRISPRCasTyper even revealed the presence of nine potential CRISPR-Cas operons (see **Supplementary Data 2**), of which two could be assigned to a Type 1-C and a Type 1-B classification system (see **Supplementary Data 2, CRISPRTyper consensus repeat sheet and Supplementary Data 4**). Furthermore, this overlap of detection confirms that these CRISPRs conform to known CRISPR-signatures.

Thus, the CRISPRs identified in the human genome fulfill the CRISPR-Cas criteria as discussed in:

- 1) David Couvin, Aude Bernheim, Claire Toffano-Nioche, Marie Touchon, Juraj Michalik, Bertrand Néron, Eduardo P C Rocha, Gilles Vergnaud, Daniel Gautheret, Christine Pourcel, CRISPRCasFinder, an update of CRISPRFinder, includes a portable version, enhanced performance and integrates search for Cas proteins, **Nucleic Acids Research, Volume 46, Issue W1, 2 July 2018, Pages W246–W251, <https://doi.org/10.1093/nar/gky425>**
- 2) Biswas A, Staals RH, Morales SE, Fineran PC, Brown CM. CRISPRDetect: A flexible algorithm to define CRISPR arrays. **BMC Genomics. 2016 May 17;17:356. doi: 10.1186/s12864-016-2627-0**
- 3) Russel J, Pinilla-Redondo R, Mayo-Muñoz D, Shah SA, Sørensen SJ. CRISPRCasTyper: Automated Identification, Annotation, and Classification of CRISPR-Cas Loci. **CRISPR J. 2020 Dec;3(6):462-469. doi: 10.1089/crispr.2020.0059.**

We thus provide significant evidence that the CRISPRs identified in the human genome fulfill the criteria as set for CRISPRs and CRISPR-associated genes. However, (as revised in the **Supplementary Discussion**) further research is needed to establish whether they also fulfill roles as shown in prokaryotes (see **Supplementary Information**).

2- Actually, the authors only show the presence of repeat sequences in the human genome, a well-known feature and they refer to reports and databases in which these repeat sequences are transcribed in certain cell lines and tumor tissues. The authors suggestion that these SRSR's can be used as biomarkers might be useful. It is recommended to rewrite the manuscript and focus on this subject.

Response:

Using RT-qPCR and the CRISPR-Cas-based SHERLOCK and DETECTR methodologies for pooled samples with malignant or normal-adjacent to tumor prostate tissues, we show differential expression patterns (see **Fig. 7 and Supplementary Fig. 5; lines 241 - 258**). These promising results reveal that at least a subset of the CRISPRs identified in the human genome can be used as putative biomarkers. However, these preliminary findings need to be validated in a follow-up study such as a retrospective, randomized and independently-validated controlled clinical trial to establish their biomarker potential.

Reviewer #2 (Remarks to the Author):

Here, the authors describe the identification in humans of a class of repeat previously thought to be limited to prokaryotes and archaea. I focused primarily on the evolutionary history and genomic distribution of SRSRs in humans as part of my review and found that I would appreciate a better exploration into the evolutionary history of such elements in the human genome. In particular, how do the authors suggest that these elements arose in the human genome and, in their eyes, gained a function? I have several comments that I believe the authors should address:

Major Concerns:

The authors present no reasoning for why SRSRs were previously thought to not exist in Eukaryotic genomes, both in the context of the Introduction and in the context of their own work as part of the Results/Discussion. Further to this, are SRSRs identified by this study conserved among closely related Eukaryotes (i.e. Great Apes) or more distantly related species (mouse comes to mind)? When ascribing function to something as evolutionarily ancient as SRSRs, it is unlikely that this functionality arose completely *de novo* in the human species.

Response:

We added this reasoning into the revised manuscript (see Supplementary Note 2 and Supplementary Discussion) and provide evidence that in more closely related Eukaryotes and more distantly related species some of the SRSRs in the revised version renamed as CRISPRs are conserved and agree with the reviewer that these elements are not only *de novo* in the human genome (see Supplementary Data 2 and 4 and Supplementary Fig. 3 and the Supplementary Information file.

What is the sequence homology of the repeat sequences of identified SRSRs? Do you identify families of SRSRs in humans?

Response:

In the revised version we present data showing that we also identified CRISPRs in the human genome belonging to known prokaryotic CRISPR repeat families or known CRISPR-Cas systems. During our analyses we did discover new repeat families, but also showed by using CRISPRMap and CRISPRloci that specific CRISPR repeat motifs, sequence families and superclasses exists in the CRISPRs as identified in the human genome (see Supplementary Data 2)

Overall, I find the manuscript lacking for statistics when terms like "enrichment" and "substantial" are used. Many of the proportions presented are out of context without a null expectation (i.e. 41.6% with polyA recognition sites seems high, but the most common poly(A) sequence homology is AATAAA and as such is likely to be fairly common in the human genome). In particular, it is unclear to me if SRSRs overlap with particular consequences/genomic features/binding sites more than expected by chance.

It would be helpful to update Figures 2/3 with such an analysis. Specifically, the authors state "we noticed an enrichment for RNA Polymerase II Subunit A (POLR2A) and the CCCTC-Binding factor (CTCF)" but do not provide any sort of statistic to show such an enrichment.

Further to this, I have no context for the number of SRSR elements that should be found to be expressed in any of the GEO databases cited in the study.

Response:

We are grateful that the reviewer made us aware of this fact and addressed this in the revised manuscript (see lines 144-149 page 7). Added text: "This analysis demonstrated that 4,126 of the 12,572 hCRISPRs (33%) overlap with one or more TFBS positions (see Fig. 3c and Supplementary Data 1 and 5), 2,668 (21%, p -value < 0.001, Z-score: 4.9507) with DNase I hypersensitive sites (see Fig. 3d and Supplementary Data 1 and 5) and 104 (1%, p -value: 0.15, Z-score: 1.1334) with CpG methylated islands (see Fig. 3d and Supplementary Data 1 and 5). Interestingly, we noticed an enrichment for footprints of POLR2A (5.5%, p -value < 0.001, Z-score: 7.079) that overlap with 696 hCRISPRs (see Fig. 3c and Supplementary Data 1 and 5)".

It is unclear to me whether SRSRs are expressed independently of the transcripts within which they are contained. Is the differential expression you observe in cancer genomes simply due to expression of such transcripts? The author's statement about EPCATs suggests this to be true and as such what is the actual diagnostic potential of SRSRs beyond EPCATs?

Response:

We addressed this question by showing with the RT-qPCR technology that the elements are expressed as independent transcript comparable in size to sncRNAs, with transcripts ranging in size of 20-50 bp (see Fig. 7 and Supplementary Fig. 5). The actual diagnostic potential of these elements beyond prostate cancer needs to be further uncovered, for which this manuscript provides the basis to do so. Our newly added RT-qPCR, DETECTR and SHERLOCK technologies demonstrate (see Figure 7 and Supplementary Fig. 5; lines 241 - 258) that this can be further explored in the near future for a wide variety of diseases as revealed in Supplementary Data 11.

Do you have any molecular evidence to show that SRSRs are expressed as distinct molecules (e.g. Northern Blotting/RT-PCR)?

Response:

We have used the RT-qPCR technology to confirm their expression as distinct molecules (see Fig. 7 and Supplementary Fig. 5).

While SRSRs may overlap with peaks for TFBS', does the actual spacer/repeat sequence harbour any homology to any known transcription factor?

Response:

Yes, some SRSRs renamed to CRISPRs have a 100% identity to for example to SP1, AP1, ATF, c-Myc and Oct-1 and includes both repeats or spacers. More examples are listed in **Supplementary Data 1** and **5**. See another example below as visualized in IVG. In red a CRISPR is visualized that resides in the human genome. In grey one sees the transcription factor binding site for Pol2, which also resides in a DNase I clusters (light purple lines) indicating that from this position active transcription is possible.

The statement in the discussion "Of main importance is our discovery that their expression enables 256 the differentiation between healthy individuals and diseased patients" does not appear to be supported by anything beyond circumstantial evidence. Can the authors show that a predictive model using a test/training regimen is predictive and that it is more predictive than the literature cited in the manuscript (i.e. Böttcher, R. et al.)?

Response:

We agree with this remark of the reviewer, but the revised manuscript now harbors proof on this matter by applying the RT-qPCR technology that revealed differential expression of two human CRISPR RNAs in pooled samples separating normal-adjacent to tumor from malignant prostate tissue (see Fig. 7, and Supplementary Fig. 5). In that respect we understand the suggestion of this reviewer to apply a predictive model, but we feel that a more rigorous approach is needed. We therefore suggest that new studies should address the biomarker potential of these elements in specific diseases in which a retrospective trial is followed by a randomized controlled trial and then an independent validation to reveal the true potential of human CRISPRs as biomarkers.

Minor Comments:

This is more of a scientific writing style choice, but I find the use of superlative words in the manuscript like "mysterious", "remarkable", and "peculiar" as scientifically imprecise (particularly in the Introduction). For example: what does a word such as "mysterious" actually mean in the context of "researchers identified a mysterious class of repeats" (L. 52)? Does it mean unexpected, surprising, dissimilar to other repeat classes, etc.? I would prefer the authors be more specific in their writing.

Response:

We thank the reviewer for these remarks, we checked the manuscript and where appropriate became more specific in our writing. Furthermore, the revised manuscript is edited by a native English speaker.

I am unsure why the section "Diagnostic potential of RNA transcripts..." begins with a large summary of the author's previous work. A one sentence summary with a citation would suffice.

Response:

We have adapted shortened this section in the revised manuscript as suggested.

Generally, grammar needs editing/correcting. Quick examples include: o L.71: "potentially, even via a more" potentially via a more

L. 165-6: "the SRSR elements made use investigate their expression activity in more detail." (I'm not sure what this sentence is saying)

Response:

We have revised the grammar in the manuscript and took the quick examples along.

sncRNAs should be defined when "small non-coding RNAs" is first used on L.78. It is currently defined on L. 149. The acronym is also not used in several places when it has been defined already (e.g. L.145-6, L. 199, etc.)

Response:

We took note of this remark and checked the manuscript on these minor editing issues and adapted them accordingly.

The discussion of CRISPR in the discussion seems out of context with the rest of manuscript as it is never mentioned anywhere else. It might be worth giving context in the Introduction so this is less jarring.

Response:

Based on the comments of reviewer 1 and the remark by reviewer 2 we took more care on this matter and introduce the CRISPR-Cas nomenclature already in the introduction and extensively address this in the **Supplementary Information** as well.

I feel the use of "new" in the title of the manuscript is not particularly accurate. They may be "new" to the human genome but are not "new" in an evolutionary or scientific sense. Additionally, what does "beyond" mean in this context?

Response: We adapted the title to “CRISPRs in the human genome are differentially expressed between malignant and normal-adjacent to tumor tissue”

REVIEWERS' COMMENTS:

Reviewer #2 (Remarks to the Author):

I appreciate the time and effort the authors took to revise the manuscript in accordance with my suggestions and can now recommend it for publication.